# The Effect of Sociodemographic Factors, Parity and Cervical Cancer on Antibiotic Treatment for Uncomplicated Cystitis in Women: A Nationwide Cohort Study

**DOI:** 10.3390/antibiotics10111389

**Published:** 2021-11-12

**Authors:** Filip Jansåker, Xinjun Li, Jenny Dahl Knudsen, Veronica Milos Nymberg, Kristina Sundquist

**Affiliations:** 1Center for Primary Health Care Research, Department of Clinical Sciences Malmö, Lund University, 205 02 Malmö, Sweden; xinjun.li@med.lu.se (X.L.); veronica.milos_nymberg@med.lu.se (V.M.N.); kristina.sundquist@med.lu.se (K.S.); 2Department of Clinical Microbiology, Rigshospitalet, Copenhagen University, 2100 Copenhagen, Denmark; Inge.Jenny.Dahl.Knudsen@regionh.dk; 3Department of Family Medicine and Community Health, Icahn School of Medicine at Mount Sinai, New York, NY 10029, USA; 4Department of Population Health Science and Policy, Icahn School of Medicine at Mount Sinai, New York, NY 10029, USA; 5Center for Community-Based Healthcare Research and Education (CoHRE), Department of Functional Pathology, School of Medicine, Shimane University, Matsue 690-0823, Japan

**Keywords:** antibiotics, cervical cancer, cystitis, parity, sociodemographic factors, UTI

## Abstract

Background: Uncomplicated cystitis is one of the most common reasons for antibiotic treatment in otherwise healthy women. Nationwide studies on antibiotic treatment for this infection and in relation to factors beyond the infection itself have hitherto not been available. Methods: This was a nationwide open cohort study consisting of 352,507 women aged 15–50 years with uncomplicated cystitis (2006–2018). The outcome was a redeemed antibiotic prescription within five days from the cystitis diagnosis. Logistic regression models were used to examine the relationship between the outcome and the predictor variables. Results: This study identified 192,065 redeemed treatments (54.5%). Several sociodemographic variables were associated with antibiotic treatment. For example, women with the lowest income had an odds ratio (OR) of 1.26 (95% CI 1.23–1.28) compared to those with the highest income. History of cervical cancer and high parity were also associated with lower treatment rates. Conclusion: This study presents novel factors beyond the infection which seem to affect the antibiotic treatment for uncomplicated cystitis in women. Future studies to investigate possible mechanisms are warranted in order to properly use our findings. This may help healthcare workers and planners to provide a more equal treatment plan for this common infection, which may reduce misuse of antibiotics, decrease costs and improve efforts against antibiotic resistance.

## 1. Introduction

The worldwide increase in antibiotic consumption and resistance is of great concern [1]. Several strategies have been suggested on how to minimize the misuse and overuse of antibiotics in order to avoid increasing antibiotic resistance in society [2,3,4]. One important step in this process is to identify whether factors beyond the infection itself may affect antibiotic treatment of common infections. Such knowledge can help to improve strategic programs against misuse of antibiotics and antibiotic resistance [3] and, at the same time, provide clues to achieve a more equal treatment plan for patients with common infections.

Urinary tract infection (UTI) is a leading cause of antibiotic use [5]. Uncomplicated cystitis, also known as uncomplicated lower UTI, is one of the most common infections in otherwise healthy women [6,7,8,9]. These infections are generally caused by the overgrowth of bacteria in the urinary tract, mainly *Escherichia coli* followed by *Staphylococcus saprophyticus,* and are thus often effectively treated with a short duration of antibiotic therapy [10,11,12,13,14,15]. Antibiotic prescription rates for signs and symptoms of UTI vary between 59 and 74% in Sweden [16,17] and 56 and 95% around Europe [8,18,19]. In general, the vast majority of all types of antibiotic prescriptions seem to be redeemed by patients in the USA [20]. However, for cystitis, only 63% reported taking their redeemed prescription with full compliance in the UK [8]. This is not surprising as uncomplicated cystitis could be regarded as self-limited in many cases, as one in four patients treated with placebo and many cases do not require antibiotic treatment [10,11]. As some patients with cystitis do not need antibiotics, the recent Swedish guidelines (2017) concluded to not recommend antibiotic prescriptions for uncomplicated cystitis with mild symptoms [21].

Sweden’s tax-financed healthcare services have the goal to provide healthcare on equal terms for the entire population, irrespective of sociodemographic factors or previous ailments. However, we have recently found that sociodemographic factors are associated with uncomplicated cystitis in primary healthcare settings [9]. Sociodemographic factors were also associated with redeemed antibiotic prescriptions in a nationwide Swedish sample where specific infectious diagnoses were not considered [22]. However, no study has yet investigated antibiotic treatment for uncomplicated cystitis in relation to a comprehensive set of sociodemographic variables, parity and history of cervical cancer, one of the most common urogenital malignancies in women. The gap in previous research is most likely due to the lack of nationwide population-based datasets from primary healthcare settings, where most UTIs are managed. Sweden is an ideal setting for such a study due to its universal healthcare system with highly complete population-based databases, and a highly diverse population in terms of origin and socioeconomic status. Thus, with this study, we aimed to investigate if factors most likely beyond the infection itself affect antibiotic treatment in women with uncomplicated cystitis.

## 2. Results

Table 1 describes the study population that consisted of 352,507 women aged 15–50 years at the time of first diagnosis of uncomplicated cystitis during the study period, 2006–2018. Of these, 192,065 (54.5%) had redeemed an antibiotic prescription within five days after the diagnosis. Treatment rates varied between 46.5% in women living in northern Sweden and 61.5% in women originating from Asia (excluding the Middle East) and Oceania. Women of low age, with low income and living in large cities tended to have higher treatment rates. These rates also tended to vary between women originating from different regions in the worlds. Women with high parity and a history of cervical cancer seemed to have lower treatment rates.

Table 2 presents the three models of the associations between the individual sociodemographic variables and antibiotic treatment (i.e., a redeemed antibiotic prescription) within five days of an uncomplicated cystitis diagnosis. In Model 1, the univariate analysis, young age, low income and four of the six immigrant groups had higher odds of antibiotic treatment compared to their corresponding reference. For example, the lowest income quartile yielded an OR of 1.23 (95% CI 1.21–1.25). Both women originating from Western countries and Asia (excluding the Middle East) and Oceania had a significantly higher OR of 1.37 compared to Swedish-born women. Not living in large cities, high parity and cervical cancer were associated with lower odds of antibiotic treatment compared to the corresponding reference. For example, women living in northern Sweden had an OR of 0.64 (95% CI 0.62–0.65) compared to those living in large cities, and women with a history of cervical cancer had an OR of 0.89 (95% CI 0.86–0.92) compared to women without a history of cervical cancer. In Model 2 that was adjusted for age, most of the results for the sociodemographic variables remained almost unchanged. In Model 3, adjusted for all covariates, there were only slight changes in the results compared with the two previous models. Both socioeconomic variables, education and income, were associated with antibiotic treatment. The lowest education level had an OR of 0.92 (95% CI 0.90–0.94) compared to the highest education level, and the lowest income level had an OR of 1.26 (95% CI 1.23–1.28) compared to the highest income level. For country of origin, women originating from Western countries, and Asia (excluding the Middle East) and Oceania had increased odds of antibiotic treatment compared to those originating from Sweden, whereas women originating from the Middle East/North Africa (MENA) had decreased odds. For example, Western women and women from Asia (excluding the Middle East) and Oceania had the same OR of 1.28 (95% CI 1.24–1.35), while women from MENA had an OR of 0.87 (95% CI 0.85–0.90) compared to women born in Sweden. The associations with cervical cancer and parity remained. For example, multipara women had an OR of 0.85 (95% CI 0.83–0.56) compared to nullipara women.

## 3. Discussion

This study found that antibiotic treatment rates for uncomplicated cystitis varied amongst different sociodemographic groups of women. Most noteworthy were the associations with country of origin, region of residence and socioeconomic status. Parity and cervical cancer also seemed to have an impact on the treatment rates. This suggests that antibiotic treatment for uncomplicated cystitis could be affected by factors that may be beyond the infection itself.

To our knowledge, worldwide, this is the first population-based study to investigate antibiotic treatment for uncomplicated cystitis in women diagnosed in primary healthcare settings and specifically the association with sociodemographic factors. The previous absence of large-scale studies in this field is likely due to the hitherto lack of nationwide primary healthcare diagnoses. Sociodemographic disparities have, however, been found in general antibiotic prescription in Sweden [22], which similarly measured “bought [redeemed] antibiotics in pharmacies”, and as the vast majority of antibiotic prescriptions seem to be redeemed by patients [20], this seems to be a good estimate of antibiotic prescriptions in general. However, the findings in that study differed in some aspects from the findings of our study. The differences can most likely be explained by the difference in the definition of the study population. Foremost, the authors stated that they did not have access to the diagnoses that led to the redeemed prescriptions [22]. Nevertheless, the findings require further investigations on prescriptions linked to clinical diagnoses. For example, foreign origin was associated with decreased odds of antibiotic prescriptions for UTI. Similarly, we found that MENA origin was associated with slightly decreased odds of antibiotic treatment compared to Swedish-born women, although other foreign groups had slightly to modestly increased odds. Other findings such as that for region of residence were similar to the findings of our study and in concordance with the raw antibiotic surveillance data of lower rates of antibiotic prescriptions in general [5].

Several explanations might exist behind the findings of our study. In general, the differences in antibiotic treatment between groups could partly be attributed to differences in healthcare access and quality between healthcare providers and demographic regions. It is also possible that certain groups of women have lower adherence rates to redeem antibiotic prescriptions than others for several reasons. For example, the distance to pharmacies could affect the willingness in a patient to travel to and thus redeem an antibiotic prescription (lack of adherence). Another reason might be age differences and cultural differences concerning the patients’ expectations to treatment options for their symptoms. For example, the somewhat higher treatment rate in young women could be due to less experience or knowledge in self-treatment compared to older women, thus increasing their desire to receive antibiotic therapy for their symptoms. Furthermore, as has been suggested before [22], it is also possible that women of foreign origin could be traveling to their countries of origin where antibiotics are more easily accessed and sometimes import these antibiotics to their new residential country for future use. Therefore, certain groups of immigrant women might not redeem antibiotics prescribed for their infection (lack of adherence) while risking mistreating their infection with a potentially inadequate antibiotic they have at home. Moreover, we have recently observed that sociodemographic differences in uncomplicated cystitis seem to exist in largely the same population of women in Sweden [9]. However, the findings of that study [9], albeit not entirely comparable, were not consistent with the present study, and they indicate that mechanisms involved in uncertainty of diagnosis and variations in patients’ expectation could be at play. For example, both women from MENA and women with higher parity had lower odds of antibiotic treatment for uncomplicated cystitis in the present study but had higher risks of uncomplicated cystitis in a recent study by our group (compared to Swedish-born women and women with no children, respectively), while the opposite was found in women from Asia (excluding the Middle East) and Oceania, who had an increased risk of uncomplicated cystitis. This inconsistency merits attention as it cannot be explained on a medical basis alone. Even though certain groups of women could have genetic [23] and microbiota [24] predisposition to uncomplicated cystitis, the treatment of this infection should not be influenced by sociodemographic factors. Instead, several other explanations beyond the infection itself might exist. For example, several other illnesses are known to present with cystitis-equivalent symptoms [25,26], and variations in prescription rates have been attributed to uncertainty in diagnostic testing in smaller observational studies [18]. The findings of lower odds of antibiotic treatment for uncomplicated cystitis in women with a history of cervical cancer or high parity support this theory, as these women most likely suffer from a greater urogenital morbidity in general, which could cause symptoms common with cystitis but also other urogenital diseases [25,26]. It is therefore possible that certain groups of immigrant women might be more exposed to diagnostic uncertainty or even misdiagnosis for various reasons (e.g., language barriers, non-conventional presentation of symptoms compared to the general population). If so, physicians might be more insecure about the diagnosis in certain sociodemographic groups of patients, which could affect their prescription of antibiotics.

An important limitation with the present study is that it did not have access to data on the symptoms or clinical presentation in the patients. Thus, it was not possible to assess whether the observed differences in antibiotic treatment depended on differences in the patients’ symptoms. Secondly, we did not have access to data on adherence rates. Although it was shown that almost all antibiotic prescriptions were redeemed by patients in a CDC report from 2017 [20], it is important to keep in mind that our study did not have access to the proportion of antibiotic prescriptions that were not redeemed (i.e., lack of adherence). Consequently, no firm conclusion can be drawn on whether the differences in antibiotic treatment observed in this study are due to differences in prescription rates (unequal healthcare) or differences in pick-up rates (differences in adherence). Further studies are needed to clarify this. Furthermore, we only had access to seven-level ATC codes for the first eight years. However, the prescription data at five-level (e.g., J01CA) were almost exclusively from UTI-specific antibiotics in those years when seven-level ATC coding was available (e.g., J01CA08; pivmecillinam). Finally, as antibiotic treatment was defined as redeemed prescriptions within five days after a diagnosis of uncomplicated cystitis, it is possible that some women received antibiotic treatment in other ways, potentially affecting the overall treatment rates. That said, considering the large size of the study population and the comprehensive nature of our data, the limitations were likely balanced by the strengths. For example, pivmecillinam and nitrofurantoin were the two most used antibiotics, and the proportion of these increased over time as the proportions of fluoroquinolones and trimethoprim decreased. These trends (data not shown) are in accordance with the national guidelines [21], smaller local studies [16,17] and the national trend of redeemed (not linked to diagnoses) UTI antibiotics in women [5], which suggests high data validity. Other major strengths include that this study involved several highly validated nationwide patient and population data registries [27,28]. In addition, our group had access to nationwide primary healthcare data with almost full coverage, which is quite unique and makes this study the first to link antibiotic prescriptions to clinical diagnoses from primary healthcare settings on a nationwide basis. Finally, the annual prescription patterns were comparable to those of the same time period in smaller local studies [16,17] as well as the overall trend on UTI antibiotics in Sweden [5], and the prescription rate in women living in large cities was similar to that of another study from a large Swedish city [16]. The consistency between our results and these previous studies supports that our data sources are robust and representative in the identification of UTI and antibiotic treatment for UTI.

Antibiotics are important drugs that save many lives and lower morbidity due to infections [29]. Although physicians, now more than ever, need to keep in mind the growing threat of antimicrobial resistance [1,2,4] and other risks [10,15,30,31] of antibiotic use when considering antibiotic treatment for their patients, omitting antibiotics when they are needed is not in accordance with medical practice. Even though uncomplicated cystitis, in many cases, can be regarded as self-limited without any evident risk of pyelonephritis [10,11], pyelonephritis could be occurring more frequent in patients treated with analgesics [12]. In addition, some groups seem to be at higher risk of both uncomplicated cystitis [9] and pyelonephritis [32], such as MENA or multipara women, who also seem to receive less antibiotic treatment for the former condition. On the other hand, the risks of uncomplicated cystitis [9] and pyelonephritis [32] seem to be lower in women living outside large cities, who also receive less antibiotics for the former. Although the clinical impact of these findings is unclear, it is important to avoid misuse and overuse of antibiotics, mainly because they drive antimicrobial resistance forward [1,2,4] and cause unwanted side effects [10,15,30,31].

Mainly thanks to the Swedish strategic program against antibiotic resistance (called “Strama” [3] in Swedish, which means to “tighten” [antibiotic prescriptions]), Sweden has one of the lowest prescription rates of antibiotics in the world [3,5]. This could explain why the general antibiotic treatment rate (54.5%) in this study was at the lower end compared to previous studies in Europe (54% to around 90%) [8,16,17,18,19]. Nevertheless, while the antibiotic prescription rates have kept decreasing in Sweden in recent years (in contrast to the rest of the world) [1], antibiotics for UTI still remain, in general, a leading cause of prescribed antibiotics in primary healthcare and have not decreased considerably over the last decade [5]. Therefore, as our results also indicate, more efforts are needed to find an adequate treatment for UTI. One necessary and important factor in these efforts is to increase the scientific knowledge in clinical decision making. From a public health perspective, healthcare planners could provide tailored educational interventions to certain sociodemographic groups in accordance with Strama’s education and awareness strategies [3]. Such programs could lead to a more equal treatment of patients and further minimize unnecessary use of antibiotics [1,2]. Further studies are needed to examine possible mechanisms behind the found associations with sociodemographic factors, parity, and cervical cancer.

## 4. Material and Methods

### 4.1. Study Design and Setting

This study was designed as an open population-based cohort study. The study population consisted of 352,507 women aged 15–50 years at the time of being diagnosed with a first diagnosis of uncomplicated cystitis during the study period from 1 January 2006 to 31 December 2018. The time period was defined according to when full nationwide prescription data were available for the study. Each woman could only be included once. The outcome variable (redeemed antibiotic prescription) was defined as the first UTI antibiotic redeemed (measured as prescription from a physician dispensed at a pharmacy) within five days after a diagnosis of uncomplicated cystitis. The outcome is generally referred to as “antibiotic treatment” in the manuscript. The STROBE statement for cohort studies was considered [33]. This research was conducted at the Center for Primary Health Care Research at Lund University and Region Skåne, Malmö, Sweden.

### 4.2. Ascertainment of Study Population

The first diagnosis of acute uncomplicated cystitis, hereafter referred to as uncomplicated cystitis, in women was identified in the Swedish Primary Health Care Register during the study period. The data sources were similar to those used in a previous study that investigated the risk of uncomplicated cystitis [9]. The 10th revision of the International Classification of Diseases (ICD-10) was used to identify cases of cystitis (i.e., ICD-10: N30) [9]. The following diagnoses were not considered as these could be suggestive of not being consistent with the diagnosis of uncomplicated cystitis [7,25,26,34]: (i) diagnoses that were specifically classified as not being acute infective cystitis (i.e., ICD-10: N301-304, and N308); (ii) history of HIV, non-male urological neoplasms, immunodeficiency disorder, diabetes, paraplegic syndrome, chronic nephritic syndrome, hereditary nephropathy, chronic pyelonephritis, hydronephrosis, nephropathy, other serious kidney diseases, kidney failure, urolithiasis, neurological bladder dysfunction, congenital or other diseases of the kidney, bladder or urinary tract, defined as any such diagnosis (ICD-10: B20-24, C64-68, D41, D80-89, E10-11, M623, N03, N07, N11, N13-23, N25-29, N32, Q60-64) within two years prior to the cystitis diagnosis; (iii) history of redeemed prescription on anti-neoplastic and/or immunomodulating agents or corticosteroids for systemic use (Anatomic Therapeutic Chemical (ATC) Classification System codes: L or H02) within six months prior to the cystitis diagnosis; or (iv) ongoing pregnancy at the time of the cystitis diagnosis.

### 4.3. Ascertainment of Outcome Variable

The following antibiotic groups were assessed based on their ATC code: penicillins with an extended spectrum (J01CA), for example, pivmecillinam; beta-lactam/b-lactam inhibitor combinations (J01CR); cephalosporins (J01DB-E,I); trimethoprim derivatives (J01EA), i.e., trimethoprim; sulphonamides and trimethoprim combinations (J01EE), i.e., trimethoprim/sulphamethoxazole; fluoroquinolones (J01MA), for example, ciprofloxacin; and nitrofuran derivatives (J01XE), i.e., nitrofurantoin. Five-level ATC codes were available for the whole period, and seven-level ATC codes were available during the first eight years (2006–2013) of the study period. The five-level ATC codes were used to include the potential antibiotic alternatives for uncomplicated cystitis listed in international [31] and/or Swedish guidelines [22] including pivmecillinam, amoxicillin/clavulanic acid (a beta-lactam/b-lactam inhibitor combination), various cephalosporines, trimethoprim with/without a sulphonamide, fluoroquinolones, nitrofurantoin and fosfomycin. Only oral antibiotic prescriptions were assessed. Fosfomycin or mono-sulphonamide antibiotics were excluded as these antibiotics were not widely available in Sweden at the time of the study (only a few cases were found in the preliminary searches of the data). The study did not consider antibiotics not recommended [21,31] to be used for uncomplicated cystitis or generally prescribed (e.g., doxycycline, metronidazole); these antibiotics were found to be used in other studies but to a very small degree [17,18].

### 4.4. Ascertainment of Predictor Variables

The predictors investigated were measured at baseline, i.e., time of the cystitis diagnosis. Age groups were defined as being between 15 and 24, 25 and 34, 35 and 44 or 45 and 50 years of age. Socioeconomic status was defined by individual education and family income. Educational level was classified into three different categories based on the duration of school years attended: compulsory schooling or less (≤9 years); short or partially completed high school education (10–11 years); or completed high school education or more, such as university or college education (≥12 years). For those aged 15–17 years in the youngest age group, the highest educational level of the parents was used. Family income was categorized into four groups based on a weighted average income in each family [9]: low (lowest income quartile of the study population), middle low/middle high (second/third quartiles) and high (highest quartile). Region of residence was grouped as large cities, southern Sweden and northern Sweden. Country of origin was categorized as originating from any of the following countries/regions: Sweden; Eastern European countries; Western countries; the Middle East/North Africa (MENA); Africa (excluding North Africa); Asia (excluding the Middle East) and Oceania; or Latin America/the Caribbean. Countries with geographical proximity and/or cultural and economic similarities were categorized together. Both first and second generations of immigrants were included in country groups other than Sweden. The categories for this study were based on the definition used in a previous study of ours [9]. Cervical cancer (Yes/No) was defined from the Swedish Cancer Register according to ICD-7 171, during the study period. Parity was defined from the Swedish Medical Birth Register and categorized as no child (nullipara), one or two children or more than two children (multipara).

### 4.5. Data Sources

The study population was identified in the Swedish Primary Healthcare Register, which includes nationwide clinical diagnoses from primary healthcare consultations in Sweden. The coverage of the data varied by time period and region. The starting time points differed because the patient records were digitalized in different time periods in the different regions. The register contained 72% of the population in Sweden in 2015 and around 90% of the population at the end of the study [9,32]. The following data registers were used to identify the outcome, parity and cervical cancer, as well as comorbidities or other complicating factors not aligned with uncomplicated cystitis: the Swedish Cancer Register, which includes data on cancer diagnosis in Sweden; the Medical Birth Register, which includes data on, e.g., pregnancies in Sweden; the Hospital Discharge Register and the Outpatient Register, which include hospital discharge diagnoses and diagnoses from outpatient specialist care, respectively; the Cause of Death Register; and The Swedish Prescribed Drug Register, which contains the specific ATC codes on redeemed drug prescriptions from all pharmacies in Sweden. The Total Population Register was used to collect data on age, country of origin, income, education and region of residence. The population registers are close to 100% complete for the Swedish population [28]. All linkages between the individual-level data in the databases were performed using a pseudonymized version of the unique 10-digit personal identification number (assigned to each person living in Sweden).

### 4.6. Statistical Analysis

Descriptive statistics (study population size, numbers and rates of antibiotic treatments) were calculated in each category of the different variables. To test for the association between the predictor variables and the outcome (redeemed antibiotic prescription for uncomplicated cystitis), logistic regression models were used to estimate odds ratios (OR) and 95% confidence intervals (CI). The study period started on 1 January 2006 and proceeded until redeemed antibiotic prescription (within five days), death, emigration or end of the study period on 31 December 2018. Three models were used, where Model 1 was a univariate model, Model 2 was adjusted for age and Model 3 was adjusted for age and the other covariates. Missing values (range 0.0–1.3%) were not excluded. For education (0.9%) and income (1.3%), they were instead included in the groups with the lowest levels of education and income, respectively. Unknown region of residence (0.7%) was included in the category large cities. Unknown country of origin (in total, 26 individuals, 0.0%) was included in the category Sweden. A two-tailed p-value of <0.05 was considered for statistical significance. We used SAS version 9.4 (SAS Institute Inc., Cary, NC, USA) for all statistical analyses.

### 4.7. Ethical Consideration

The present study was a non-intervention nationwide register study of pseudonymized secondary data obtained from Swedish authorities and was approved by the ethical review board in Lund. All methods were performed in accordance with the relevant guidelines and regulations.

### 4.8. Role of Funding Source

The funding sources of this study were all non-commercial and had no role in the study design; the collection, analysis, and interpretation of data; the writing of the report; or the decision to submit the paper for publication. All authors confirm that they had full access to all the data in the study and accept responsibility to submit the paper for publication.

## 5. Conclusions

This study found that certain sociodemographic factors, parity, and cervical cancer might affect antibiotic treatment for uncomplicated cystitis in women. These findings represent important new information on factors beyond the infection itself that might affect the treatment of this bacterial infection in women. Future studies to investigate possible mechanisms are warranted in order to properly use our findings. The findings may help to provide a more equal treatment plan for this common infection, which may spare women from misuse of antibiotics and decrease the costs for both patients and the healthcare system. Strategic programs could also use our findings in efforts against antibiotic resistance.

## Figures and Tables

**Table 1 antibiotics-10-01389-t001:** The study population of women with uncomplicated cystitis and the number of antibiotic treatments.

	Total Population		Redeemed Antibiotics		Pick-Up Rate
	No.	%	No.	%	%
**Age groups (years)**					
15–24	85,892	24.4	49,519	25.8	57.7
25–34	107,447	30.5	57,915	30.2	53.9
34–44	100,341	28.5	53,484	27.8	53.3
45–50	58,827	16.7	31,147	16.2	52.9
**Educational level**					
≤9	51,464	14.6	27,873	14.5	54.2
10–11	57,802	16.4	31,699	16.5	54.8
≥12	243,241	69.0	132,493	69.0	54.5
**Family income**					
Low	88,113	25.0	50,608	26.3	57.4
Middle-low	87,967	25.0	48,071	25.0	54.6
Middle-high	88,272	25.0	47,253	24.6	53.5
High	88,155	25.0	46,133	24.0	52.3
**Region of** **residence**					
Large cities	237,810	67.5	137,075	71.4	57.6
Southern Sweden	82,942	23.5	40,238	21.0	48.5
Northern Sweden	31,755	9.0	14,752	7.7	46.5
**Country of origin**					
Sweden	266,071	75.5	142,963	74.4	53.7
Eastern Europe	20,167	5.7	11,380	5.9	56.4
Western countries	13,164	3.7	8080	4.2	61.4
Middle East/North Africa	25,591	7.3	13,638	7.1	53.3
Africa (excluding North Africa)	8256	2.3	4519	2.4	54.7
Asia (excluding the Middle East) and Oceania	12,993	3.7	7987	4.2	61.5
Latin America and the Caribbean	6265	1.8	3498	1.8	55.8
**Cervical** **cancer**					
No	339,498	96.3	185,332	96.5	54.6
Yes	13 009	3.7	6733	3.5	51.8
**Parity**					
No child	102,533	29.1	58,512	30.5	57.1
One or two children	186,629	52.9	100,812	52.5	54.0
More than two children	63,345	18.0	32,741	17.0	51.7
All	352,507	100.0	192,065	100.0	54.5

**Table 2 antibiotics-10-01389-t002:** The association of individual sociodemographic factors, cervical cancer and parity with antibiotic treatment (192,065 cases) for cystitis.

Covariates	Model 1				Model 2				Model 3			
	OR	95% CI		*p-*Value	OR	95% CI		*p-*Value	OR	95% CI		*p-*Value
**Age (ref. age 45–50 years)**												
15–24	1.21	1.19	1.24	<0.0001	1.21	1.19	1.24	<0.0001	1.18	1.15	1.21	<0.0001
25–34	1.04	1.02	1.06	0.0002	1.04	1.02	1.06	0.0002	1.00	0.98	1.02	0.8216
35–44	1.01	0.99	1.04	0.1701	1.01	0.99	1.04	0.1701	0.98	0.96	1.00	0.1180
**Educational level (ref. ≥12 years)**												
≤9	0.99	0.97	1.01	0.1998	0.97	0.95	0.98	0.0003	0.92	0.90	0.94	<0.0001
10–11	1.02	1.00	1.03	0.1078	1.04	1.03	1.06	<0.0001	1.05	1.03	1.07	<0.0001
**Family income (ref. high)**												
Low	1.23	1.21	1.25	<0.0001	1.19	1.17	1.21	<0.0001	1.26	1.23	1.28	<0.0001
Middle-low	1.10	1.08	1.12	<0.0001	1.07	1.05	1.09	<0.0001	1.15	1.12	1.17	<0.0001
Middle-high	1.05	1.03	1.07	<0.0001	1.03	1.01	1.05	0.0032	1.07	1.05	1.09	<0.0001
**Region of** **residence (ref. large cities)**												
Southern Sweden	0.69	0.68	0.70	<0.0001	0.68	0.66	0.69	<0.0001	0.68	0.66	0.69	<0.0001
Northern Sweden	0.64	0.62	0.65	<0.0001	0.62	0.61	0.64	<0.0001	0.62	0.60	0.63	<0.0001
**Country of origin (ref. Sweden)**												
Eastern Europe	1.12	1.08	1.15	<0.0001	1.14	1.10	1.17	<0.0001	1.02	0.99	1.05	0.3128
Western countries	1.37	1.32	1.42	<0.0001	1.41	1.36	1.46	<0.0001	1.28	1.24	1.33	<0.0001
Middle East/North Africa	0.98	0.96	1.01	0.1786	1.00	0.97	1.02	0.8769	0.87	0.85	0.90	<0.0001
Africa (excluding North Africa)	1.04	1.00	1.09	0.0713	1.05	1.01	1.10	0.0185	0.98	0.93	1.02	0.3200
Asia (excluding the Middle East) and Oceania	1.37	1.33	1.42	<0.0001	1.41	1.36	1.46	<0.0001	1.28	1.24	1.33	<0.0001
Latin America and the Caribbean	1.09	1.04	1.15	0.001	1.12	1.07	1.18	<0.0001	0.99	0.94	1.04	0.7659
**Cervical cancer (ref. no)**	0.89	0.86	0.92	<0.0001	0.88	0.85	0.91	<0.0001	0.89	0.86	0.93	<0.0001
**Parity (ref. none)**												
One or two	0.88	0.87	0.90	<0.0001	0.92	0.90	0.93	<0.0001	0.92	0.91	0.94	<0.0001
More than two	0.81	0.79	0.82	<0.0001	0.85	0.83	0.86	<0.0001	0.85	0.83	0.86	<0.0001

Model 1: univariate model; Model 2: age-adjusted model; Model 3: fully adjusted.

## Data Availability

This study made use of several national registers and, owing to legal concerns, data cannot be made openly available. Further information regarding the health registries is available from the Swedish National Board of Health and Welfare: https://www.socialstyrelsen.se/en/statistics-and-data/registers/ (accessed on 26 October 2021), and Kristina Sundquist, senior author of this study.

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
