# Peer review of "The Effect of Sociodemographic Factors, Parity and Cervical Cancer on Antibiotic Treatment for Uncomplicated Cystitis in Women: A Nationwide Cohort Study"

_antibiotics, 2021, doi:10.3390/antibiotics10111389_

Round 1
Reviewer 1 Report
Dear Editors and authors,
This is an interesting paper that deserves to be published, having a sound and robust database. Here are my comments, questions and recommendations:
- from a structural point of view, the material and methods section should come BEFORE Discussion section (I think it has been accidently misplaced)
- while it has been adequately explained why low income and living in large cities is significantly associated with higher treatment rate, it would be interesting to develop a little bit more on why younger age (particular adolescents) are more prone to higher antibiotic treatment rate
- in the discussions, it would be interesting to elaborate a little bit more on the concept of adherence/compliance
- in the lines 123-127 it Is written: „Sociodemographic disparities have, however, been found 123 on general antibiotic prescription in Sweden.[22] That study measured “bought [re- deemed] antibiotics in pharmacies”, which can be a good estimate of antibiotic prescriptions as the vast majority of antibiotic prescriptions seems to be redeemed by patients.[20] However, the findings in that study ...”. It is not clear : the first „that study” (line 124) refers to reference 22, from previous sentence? If yes, why it has another reference at the end? (20). The second „last study” (line 127) refers to what study? I highly suggest that authors refers to eg: „ Smith et al (2014) study” or to „our/current study” for more clarity and precision.
- in the line 150: „ However, the findings of that study....” – it is unclear if „that study” refers to reference no 2 or 22.
Otherwise, congratulations!
Author Response
Comments and Suggestions for Authors
Dear Editors and authors,
This is an interesting paper that deserves to be published, having a sound and robust database. Here are my comments, questions and recommendations:
Response: We thank you for your time and comments that identified several opportunities to strengthen our work. We describe below how we have addressed these.
- from a structural point of view, the material and methods section should come BEFORE Discussion section (I think it has been accidently misplaced)
Response: Thank you for noticing! However, we have followed the journal’s author instructions and placed the Discussion section before Materials and Methods.
- while it has been adequately explained why low income and living in large cities is significantly associated with higher treatment rate, it would be interesting to develop a little bit more on why younger age (particular adolescents) are more prone to higher antibiotic treatment rate
Response: We added a paragraph in the discussion section where we have elaborated more on this (line 148-150): “For example, the somewhat higher treatment rate in young women could be due less experience or knowledge in self-treatment compared to older women thus increasing their desire to receive antibiotic therapy for their symptoms.”
- in the discussions, it would be interesting to elaborate a little bit more on the concept of adherence/compliance
Response: We considered this comment and agree with the suggestions. We have elaborated on this in the discussion section (line 143-146):” It is also possible that certain groups of women have lower adherence rate to redeem antibiotic prescriptions than others for several reasons. For example, the distance to pharmacies could affect the willingness in a patient to travel to and thus redeem an antibiotic prescription (lack of adherence).” (In addition to the already included paragraph (line 153-155): “Furthermore, as has been suggested before,[22] it is also possible that women of foreign origin could be travelling to their countries of origin where antibiotics are more easily accessed and sometimes import these antibiotics to their new residential country for future use. Therefore, certain groups of immigrant women might omit to redeem antibiotics prescribed for their infection (lack of adherence),”)
- in the lines 123-127 it Is written: „Sociodemographic disparities have, however, been found 123 on general antibiotic prescription in Sweden.[22] That study measured “bought [re- deemed] antibiotics in pharmacies”, which can be a good estimate of antibiotic prescriptions as the vast majority of antibiotic prescriptions seems to be redeemed by patients.[20] However, the findings in that study ...”. It is not clear : the first „that study” (line 124) refers to reference 22, from previous sentence? If yes, why it has another reference at the end? (20). The second „last study” (line 127) refers to what study? I highly suggest that authors refers to eg: „ Smith et al (2014) study” or to „our/current study” for more clarity and precision.
Response: Thank you for noticing this. We understand the confusion in this paragraph and have now corrected the reference numbering (line 126-130):
Sociodemographic disparities have, however, been found on general antibiotic prescription in Sweden,[22] which similarly measured “bought [redeemed] antibiotics in pharmacies, and as the vast majority of antibiotic prescriptions seems to be redeemed by patients[20] this seems like a good estimate of antibiotic prescriptions in general.
- in the line 150: „ However, the findings of that study....” – it is unclear if „that study” refers to reference no 2 or 22.
Response: Thank you for noticing this. We have added the correct reference number after „that study”.
Otherwise, congratulations!
Response: Thank you!
Reviewer 2 Report
Very interesting article. As a reader from a southern European country, an uncomplicated urinary tract infection always or nearly always results in an antibiotic prescribing. This also shared by many other readers in other parts of the world. The fact that only half of the infections were associated with an antibiotic prescription should be better explained in the paper as many readers would find this finding a little surprising. You mention in the introduction section that ’Antibiotic prescription rates for signs and symptoms of UTI vary between 59-74% in Sweden, but you only found that 54.5% of your sample had redeemed an antibiotic prescription within five days. This means that a percentage of your patients who did not redeem antibiotics might have been treated. You should discuss this variation and you might also consider a limitation of this population-based study.
Minor aspects
It is not clear to me if you followed these patients. Since only one half of the patients were treated with antibiotics it would be interesting to know if you checked for complications for example (pyelonephritis) or other contacts to the health care system.
Why did you only consider women up to 50 years? Is there a specific reason for that?
How many records do you have in this database? This should be mentioned in the paper.
What does MENA stand for should be mentioned in the text.
Considering an 'association link' between income, migration, etc. and antibiotic therapy for patients with uncomplicated UTIs as stated throughout the manuscript should be mentioned more cautiously. You should tone down this 'association' and describing this as a correlation rather than an association would be more appropriate.
Author Response
Response: Thank you for your time and comments that identified several opportunities to strengthen our work. We describe below how we have addressed the comments. Please see our responses below.
Very interesting article. As a reader from a southern European country, an uncomplicated urinary tract infection always or nearly always results in an antibiotic prescribing. This also shared by many other readers in other parts of the world. The fact that only half of the infections were associated with an antibiotic prescription should be better explained in the paper as many readers would find this finding a little surprising. You mention in the introduction section that ’Antibiotic prescription rates for signs and symptoms of UTI vary between 59-74% in Sweden, but you only found that 54.5% of your sample had redeemed an antibiotic prescription within five days. This means that a percentage of your patients who did not redeem antibiotics might have been treated. You should discuss this variation and you might also consider a limitation of this population-based study.
Response: Thank you for this important suggestion. We have discussed this in the manuscript (please see below). In Sweden, antibiotic prescription is not recommended to uncomplicated cystitis unless there are severely bothersome symptoms for the patients. The prescriptions rates at group-level (e.g. in large cites, 57.6%) were similar to those (ref 16) from a large city in Sweden (59%). Certain groups of women even had higher prescriptions rates, e.g. women from Western and Asian origin (>61%)
(line 195-198) … Finally, as antibiotic treatment was defined as redeemed prescriptions within five days after a diagnosis of uncomplicated cystitis it is possible that some women got antibiotic treatment in other ways, potentially affecting the overall treatment rates.
(line 211-213)… the prescription rate in women living in large cities was similar to that of another study from a large Swedish city.[16]
(line 205-309)… The study did not consider antibiotics not recommended[27,32] to be used for uncomplicated cystitis or generally prescribed (e.g., doxycycline, metronidazole etc.); these antibiotics have been found to be used in other studies but to a very small degree.[17,18]
Minor aspects
It is not clear to me if you followed these patients. Since only one half of the patients were treated with antibiotics it would be interesting to know if you checked for complications for example (pyelonephritis) or other contacts to the health care system.
Response: This is a good suggestion that deserves a study of its own. In the present study, we did not consider complications.
Why did you only consider women up to 50 years? Is there a specific reason for that?
Response: Our focus was not on post-menopausal women but rather on fertile women. In addition, there is a discussion on whether post-menopausal women could be regarded as having uncomplicated cystitis, i.e., the outcome of the present study.
How many records do you have in this database? This should be mentioned in the paper.
Response: Please see the subheading “Data sources” in the methods section where this is mentioned.
What does MENA stand for should be mentioned in the text.
Response: Thank you for noticing this. MENA is an abbreviation commonly used for the region Middle East/North Africa. This was already mentioned under abbreviations and in the methods (line 317), but we have now added the full name on line 105 when it is first mentioned.
Considering an 'association link' between income, migration, etc. and antibiotic therapy for patients with uncomplicated UTIs as stated throughout the manuscript should be mentioned more cautiously. You should tone down this 'association' and describing this as a correlation rather than an association would be more appropriate.
Response: Thank you for this comment. We have considered this carefully but found, after discussion with a very experienced epidemiologist and statistician, that “association” is more appropriate than “correlation”, the latter being more specific. Not all associations are correlations and, in the present study, the more general term “association” seemed therefore to be more appropriate. In addition, neither “correlation” nor “association” implies causation.
Reviewer 3 Report
Jansåker et al. explored the effect of sociodemographic factors, parity, and cervical cancer on antibiotic treatment for uncomplicated cystitis in women in a nationwide cohort study in Sweden. Despite the fact that the study included more than 350000 participants, making it very representative, I see no real value of this study for the scientific community. Namely, the authors performed relatively cursory statistical analysis (probably owing to the fact that their registry had insufficient amount of relevant data) which prevented making any valuable inferences. The authors state: “The findings may help healthcare workers and planners to provide a more equal treatment plan for women, while helping to reduce unnecessary antibiotic use and resistance in the society”, yet these conclusions are not supported by the results.
In addition, the authors properly addressed all the limitations present in the manuscript, yet some of those limitations simply reduce the value of the study too much. Specifically, as the authors said themselves, from current results it cannot be concluded whether the differences in antibiotic treatment observed in this study are due to differences in prescription rates (unequal healthcare) or differences in pick-up rates (differences in adherence). This comparison should be the most important aspect of studies as such.
Author Response
Jansåker et al. explored the effect of sociodemographic factors, parity, and cervical cancer on antibiotic treatment for uncomplicated cystitis in women in a nationwide cohort study in Sweden. Despite the fact that the study included more than 350000 participants, making it very representative, I see no real value of this study for the scientific community. Namely, the authors performed relatively cursory statistical analysis (probably owing to the fact that their registry had insufficient amount of relevant data) which prevented making any valuable inferences. The authors state: “The findings may help healthcare workers and planners to provide a more equal treatment plan for women, while helping to reduce unnecessary antibiotic use and resistance in the society”, yet these conclusions are not supported by the results.
Response: Thank you for your time in reviewing and commenting on this nationwide register study that investigates if the antibiotic treatment rates for uncomplicated cystitis were affected by factors likely beyond the infection itself.
We believe that the present study adds important knowledge to the scientific community as no earlier studies, to the best of our knowledge, have used nationwide primary healthcare data (diagnoses) linked to redeemed antibiotic prescriptions. The study identifies several factors most likely beyond the infection itself that are associated with variation in antibiotic treatment rates and was based on several highly validated nationwide registers with comprehensive sociodemographic and healthcare data.
However, our enthusiasm for our study should not preclude us from also acknowledging its limitations. We have therefore toned down the conclusions in both the abstract and the discussion, where we mention that future studied are needed to examine mechanisms in order to use our findings in a proper manner. The following sentences at the end of the abstract are revised:
(line 28-31)… “…Future studies to investigate possible mechanisms are warranted in order to properly use our findings. This may help healthcare workers and planners to provide a more equal treatment plan for this common infection, which may reduce misuse of antibiotics, decrease costs and improve efforts against antibiotic resistance.”
In addition, the authors properly addressed all the limitations present in the manuscript, yet some of those limitations simply reduce the value of the study too much. Specifically, as the authors said themselves, from current results it cannot be concluded whether the differences in antibiotic treatment observed in this study are due to differences in prescription rates (unequal healthcare) or differences in pick-up rates (differences in adherence). This comparison should be the most important aspect of studies as such.
Response: Thank you for commenting on this limitation, which we have mentioned. We agree and think that this is a limitation that could be examined in future, smaller studies. Please see above how we have revised our conclusions. We have also included this comment in a new sentence in the limitations:
(line 190-192)… “….due to differences in prescription rates (unequal healthcare) or differences in pick-up rates (differences in adherence). Further studies are needed to clarify this.”
Round 2
Reviewer 3 Report
No comments.